# Exogenous Selenium and Biochar Application Modulate the Growth and Selenium Uptake of Medicinal Legume *Astragalus* Species

**DOI:** 10.3390/plants12101957

**Published:** 2023-05-11

**Authors:** Shengjun Ma, Guangwei Zhu, Rozi Parhat, Yuanyuan Jin, Xueshuang Wang, Wenping Wu, Wanli Xu, Yanling Wang, Wenfeng Chen

**Affiliations:** 1College of Food and Pharmaceutical Science, Xinjiang Agricultural University, Urumqi 830052, China; 2Institute of Chinese Materia Medica, China Academy of Chinese Medical Sciences, Beijing 100700, China; 3Institute of Soil Fertilizer and Agricultural Water Conservation, Xinjiang Academy of Agricultural Sciences, Urumqi 830091, China; 4State Key Laboratory of Agrobiotechnology, College of Biological Sciences and Rhizobium Research Center, China Agricultural University, Beijing 100193, China

**Keywords:** biochar application, hyperaccumulator, legumes, Se content, active components

## Abstract

*Astragalus* species have a certain capacity to enrich selenium (Se) and are the strongest Se hyperaccumulator legumes known globally at present. The biochar application to medicinal plants has been reported to affect plant metabolites. In this study, we aimed to employ hyperaccumulating *Astragalus* species in the plant growth of selenium-lacked soil, while also investigating the impact of varying selenium doses and biochar application on legumes growth, selenium content, and secondary metabolite production. Applying biochar to soil, along with a Se concentration of 6 mg/kg, significantly enhanced the growth, Se content, total polysaccharide content, and calycosin-7-glucoside content of *Astragalus* species (*p* < 0.05). Importantly, the Se and biochar application also led to a significant improvement in Se content in ABH roots (*p* < 0.05). Meanwhile, the content of total flavonoids in ABH roots could be promoted by a Se concentration of 3 mg/kg and biochar application in soil. Additionally, the Se enrichment coefficients of *Astragalus* species under Se treatments were significantly higher than those under control treatment, with a marked difference observed across all treatments, whether roots or above-ground (*p* < 0.05). Remarkably, the Se transport coefficients of *Astragalus* species were observed to be lower than one, except for the transport coefficient of AB in the Se concentration of the control treatment (0 mg/kg). This result showed that a medium concentration treatment of Se and biochar application in soil not only promotes the growth of *Astragalus* species and the uptake of exogenous Se but also increases the active component content, meanwhile enhancing the Se enrichment and transport capacity. Taken as a whole, the present findings offer a more comprehensive understanding of the interplay between distinct Se levels, as well as the addition of biochar in soil, providing valuable insight for the cultivation of Se-rich *Astragalus* in Se-deficient soil-plant systems.

## 1. Introduction

Astragali Radix, the dried roots of *Astragalus membrauceus* (Fisch.) Bge. var. *mongholicus* (Bge.) Hsiao (*ABH*) or *A. membrauaceus* (Fisch.) Bge (*AB*) is a Chinese herbal medicine which can also be consumed as a food (State Pharmacopoeia Commission, 2020). Astragali Radix contains lots of active components, such as flavonoids, polysaccharides, and a variety of trace elements like selenium (Se), exhibiting pharmacological effects [1,2]. In particular, calycoflavone-7-glucoside is a flavonoid compound, which is also the main active component and secondary metabolite of *Astragalus*. Currently, the strongest Se hyperaccumulators are species from the genus *Astragalus* of legumes [3]. *Astragalus* species can accumulate selenium (Se) to varying degrees. Moreover, notable disparities exist in the growth conditions and active component levels across different *Astragalus* species [2,4,5]. As a herbal medicine with strong Se enrichment capacity and certain health effects in itself, the market potential for the development of Se-rich *Astragalus* species is great [6]. Thus, the implementation of appropriate procedures to enhance *Astragalus* species growth and improve its ability to accumulate Se and active constituents would be advantageous for the advancement of this herbal remedy.

Selenium (Se), an essential trace element in the human body, has the functions of antioxidation, immune enhancement, and preventing various diseases. A lack of Se will lead to organ and system dysfunction, resulting in various human diseases [7,8,9]. Se cannot be synthesized in the human body spontaneously and it is also easy to be discharged from the body. Therefore, Se-rich food and related healthcare products have gradually attracted wide attention. As selenium (Se) deficiency remains ubiquitous in agricultural soil, implementing agronomic Se biofortification represents a pivotal strategy to alleviate associated risks of malnutrition and related diseases in both human and animal populations [10]. Studies showed that applying exogenous Se, such as Se fertilizer in the soil or on the leaves, could effectively improve the Se content in the plants [11,12].

There is increasing interest in soil amendments, such as biochar for the sustainable production of high-value crops and medicinal plants [13,14]. Additionally, biochar as a humus, mainly composed of carbon, hydrogen, and oxygen, can increase the utilization rate of nitrogen by plants through its strong adsorption capacity to reduce nitrogen loss, promote plant growth and accumulate active components [15,16]. Biochar obtained by heating and decomposing tree and crop residues under anoxic conditions has been widely confirmed in enhancing the soil fertility, improving soil physical and chemical properties, and increasing crop yield [15,17]. Due to the presence of pharmacologically active (secondary metabolites) compounds produced as a defense mechanism, the effectiveness of biochar in medicinal plants might differ from that in food crops. Studies have reported that biochar application could influence the metabolites of medicinal plants [18]. While the addition of biochar has been demonstrated to improve plant productivity, very few studies have explored their effects on the growth, physiology, and secondary metabolites in *Astragalus* species.

Therefore, the effects of Se and biochar application in *Astragalus* species of legumes on the growth, Se content, active component contents, and enrichment and transport capacity of Se were investigated in this study. We expect that our results and methods applied herein could provide some references for cultivating Se-rich *Astragalus* species and improving soil condition.

## 2. Results

### 2.1. Comparison of Growth Indexes of Astragalus Species in Different Se Concentrations

The Se hyperaccumulator *Astragalus* species was used to elucidate the plant response under different concentrations of Se dosed and with (without) biochar in soil. As indicated in Figure 1, the differences were inconsistent in different treatments, respectively. Overall, the plant height and main root length in *Astragalus* species in soil with the biochar application were greater than those of without the biochar application from the same Se concentrations treatments. The greatest plant height and main root length among all treatments were observed in GTXB. There were significant differences (*p* < 0.05) in plant height and main root length among GTCK, GCK, and JCK, while no significant differences (*p* > 0.05) were observed between GCK and JCK. Additionally, statistical differences (*p* < 0.05) were found in the main root length of GTXA and GTXB relative to JXA and JXB, respectively. No significant changes were observed in the remaining treatments. There were statistical differences among the soil treatments with biochar application (*p* < 0.05). The plant height and main root length of GTXB were highest (66.350 ± 2.519 cm and 16.050 ± 0.288 cm, respectively) in different Se concentrations treatments of the soil with biochar application, and that of GTCK and GTXD were the lowest (58.217 ± 2.670 cm and 12.833 ± 0.338 cm, respectively).

As indicated in Figure 2, the rhizome diameter of *Astragalus* species was found to be significantly greater (*p* < 0.05) in soil with biochar application, as compared to those without biochar application, under the same Se concentration treatments. Notably, statistically significant differences (*p* < 0.05) were observed in the rhizome diameter of soil with biochar application, under the same Se concentrations, except at the 12 mg/kg Se concentration. The ground diameter had no differences (*p* > 0.05) between the species in the same Se concentration treatments, except for in the Se concentration of 6 mg/kg, where differences were observed. There were statistical differences among the treatments in soil with biochar application (*p* < 0.05). The rhizome diameter and ground diameter of GTXB were the highest (5.840 ± 0.294 mm and 2.603 ± 0.330 mm, respectively) in different Se concentration treatments of the soil with biochar application, and that of GTCK and GTXD were the lowest (4.158 ± 0.183 mm and 1.383 ± 0.186 mm, respectively).

As indicated in Figure 3, the whole plant dry weight and root dry weight in *Astragalus* species in soil with biochar application were greater than those without biochar application in the same Se concentrations treatments. There were significant differences (*p* < 0.05) in the whole plant dry weight in the control treatment (0 mg/kg) and the Se concentration treatment of 3 mg/kg between the soil with biochar application and without biochar application, respectively. The root dry weight in the control treatment was statistically similar to the whole plant dry weight in the control treatment and the Se concentration treatment of 3 mg/kg. There were statistical differences among the treatments in the soil with biochar application (*p* < 0.05). The whole plant dry weight and root dry weight of GTXB were the highest (4.418 ± 0.303 g and 1.740 ± 0.080 g, respectively) in different Se concentration treatments of the soil with biochar application, and that of GTXD was the lowest (2.143 ± 0.177 g and 1.042 ± 0.119 g, respectively).

### 2.2. Comparison of Se Content of Astragalus Species in Different Se Concentrations

#### 2.2.1. Effects of Biochar Application on Se Content in *ABH* Roots

As shown in Figure 4, the Se content of roots in *ABH* with biochar application was higher than that of *ABH* without biochar application in the same Se concentrations. There were significant differences in Se concentrations ≥6 mg/kg (*p* < 0.05) between *ABH* with and without biochar application in *ABH* roots. The Se content in *ABH* roots first increased and then decreased with the increase of Se concentrations whether *ABH* with biochar application or *ABH* without biochar application, and which were both the greatest in Se concentration of 9 mg/kg. Contrarily, the *ABH* roots demonstrated no significant difference (*p* > 0.05) in Se concentration at 9 mg/kg compared to 12 mg/kg in the corresponding treatments. Conversely, they showed statistical variations (*p* < 0.05) within other Se concentration treatments. Among the biochar soil treatments, root samples from GTXC exhibited the highest Se content (5.708 ± 0.404 mg/kg), while those from GTCK demonstrated the lowest Se content (0.538 ± 0.009 mg/kg).

#### 2.2.2. Effects of Biochar Application on Se Content in *ABH* Above-Ground

As shown in Figure 5, the Se content in the above-ground parts of *ABH* showed an initial increase followed by a decrease as Se concentrations were elevated, regardless of whether biochar was applied to the *ABH* or not. Moreover, the Se content in the above-ground parts of *ABH* displayed a significant variation (*p* < 0.05) when contrasting the Se concentration treatments with and without biochar application, except for the 12 mg/kg treatment where no significant differences were detected. The Se content in both instances was maximum at a concentration of 9 mg/kg. In contrast, in the applied biochar treatments, roots from GTXC demonstrated the highest Se content (3.099 ± 0.042 mg/kg), while those from GTCK exhibited the lowest Se content (0.116 ± 0.004 mg/kg).

### 2.3. Comparison of Se Content in the Roots of ABH and AB

As shown in Figure 6, in general, the Se concentration in *ABH* and *AB* roots gradually increased alongside a corresponding increase in the applied Se concentration. At Se concentrations ≤6 mg/kg, the Se concentration in the roots of *ABH* was found to be significantly higher than that in AB roots subjected to the same concentration treatments. Conversely, at Se concentrations ≥9 mg/kg, the Se concentration in *ABH* roots was lower than that of *AB* roots. Among the cultivars examined, the roots of GXC cultivated under Se concentrations exhibited the highest Se concentration (4.150 ± 0.238 mg/kg), which did not significantly differ from that of GXD roots (4.069 ± 0.320 mg/kg). The lowest Se concentration was observed in GCK roots (0.317 ± 0.025 mg/kg). Notably, significant differences (*p* < 0.05) were detected in Se content among all *AB* root treatments. In *AB*, JXD exhibited the highest Se concentration (6.252 ± 0.162 mg/kg), while the lowest was observed in JCK roots (0.317 ± 0.025 mg/kg).

### 2.4. Comparison of Se Content in the above Ground of ABH and AB

As shown in Figure 7, a higher Se content was present above-ground from the same concentration treatment in *ABH* compared to *AB*, and a similar response was observed in Se content above-ground from *Astragalus* species which increased with the increase of Se concentrations. For both *ABH* and *AB*, the Se content above-ground was the highest (2.896 ± 0.080 mg/kg and 2.824 ± 0.258 mg/kg, respectively) in Se concentrations of 12 mg/kg, and the lowest (0.228 ± 0.032 mg/kg and 0.224 ± 0.014 mg/kg, respectively) in the control treatment. There were no significant differences (*p* > 0.05) in the Se content of *ABH* above-ground between GXC and GXD, while there were significant differences (*p* < 0.05) in the Se content of *AB* above-ground between JXC and JXD.

#### 2.4.1. Effects of Biochar Application on Total Flavonoid Content in *ABH* Roots

The effects of biochar application on total flavonoid content in *ABH* roots are shown in Figure 8. The content of total flavonoids in *ABH* roots increased first and then decreased with the increase of Se concentrations. The total flavonoid content in *ABH* roots with biochar application was higher than *ABH* without biochar application, and there were significant differences in Se concentrations of 9 mg/kg and 12 mg/kg (*p* < 0.05). The content of total flavonoid in *ABH* roots was the highest (0.308 ± 0.018% and 0.320 ± 0.005%, respectively) in Se concentration of 3 mg/kg whether *ABH* with biochar application or *ABH* without biochar application and the lowest (0.229 ± 0.017% and 0.288 ± 0.006%, respectively) in Se concentration of 12 mg/kg.

#### 2.4.2. Effects of Biochar Application on Total Flavonoid Content in *ABH* Roots

The effects of biochar application on total polysaccharide content in *ABH* roots are shown in Figure 9. The content of total polysaccharides in *ABH* roots increased first and then decreased with the increase of Se concentrations. The content of total polysaccharides in *ABH* roots with biochar application was higher than *ABH* without biochar application, and there were significant differences in any Se concentrations (*p* < 0.05). The content of total polysaccharide was the highest (14.150 ± 0.159% and 15.602 ± 0.366%, respectively) in *ABH* roots in Se concentration of 6 mg/kg whether *ABH* with biochar application or *ABH* without biochar application and the lowest (11.692 ± 0.211% and 12.978 ± 0.138%, respectively) in Se concentration of control treatment (0 mg/kg).

#### 2.4.3. Effects of Biochar on Calycosin-7-glucoside Content in *ABH* Roots

The effects of biochar application on calycosin-7-glucoside content in *ABH* roots are shown in Figure 10. The content of calycosin-7-glucoside in *ABH* roots increased first and then decreased with the increase of Se concentrations. The content of calycosin-7-glucoside in *ABH* roots with biochar application was higher than *ABH* without biochar application, and there were no significant differences in other Se concentrations (*p* > 0.05) except for the control treatment. The content of calycosin-7-glucoside was the highest (0.044 ± 0.006% and 0.047 ± 0.004%, respectively) in *ABH* roots in Se concentration of 6 mg/kg whether *ABH* with biochar application or *ABH* without biochar application and the lowest (0.027 ± 0.003% and 0.030 ± 0.001%, respectively) in Se concentrations of control treatment.

#### 2.4.4. Effects of Different Treatments on the Accumulation and Transport of Se in *Astragalus* Species

The results of Se enrichment and transport in *Astragalus* species in different treatments of Se concentration are shown in Table 1. The Se enrichment coefficients of *Astragalus* species in Se treatments were higher than that in the control treatment and there were significant differences (*p* < 0.05) whether roots or above-ground in all treatments. The Se transport coefficients of *Astragalus* species were less than 1, except for the transport coefficient of *AB* in the Se concentration of the control treatment.

The Se enrichment coefficients in roots of *ABH* and *ABH* with biochar application first increased and then decreased with the increase of the Se concentrations, and while in roots of *ABH* with biochar application were higher than that of the *ABH*, and the differences were significant (*p* < 0.05). The Se enrichment coefficients in *ABH* roots reached the maximum (9.169 ± 1.032 and 19.468 ± 2.557, respectively) in Se concentration of 6 mg/kg whether it is *ABH* with biochar application or *ABH* without biochar application, respectively, and the lowest (2.463 ± 0.187 and 8.414 ± 1.602, respectively) in Se concentration of control treatment. The Se enrichment coefficients in the above-ground *ABH* with biochar application were the highest in *Astragalus* species in the same treatments of Se concentrations except for the Se concentration of the control treatment. The Se enrichment coefficient in above-ground of *ABH* with biochar application was the highest (8.768 ± 0.251) in Se concentration of 9 mg/kg, followed by 6 mg/kg (8.516 ± 0.302), but there were no significant differences (*p* > 0.05) between Se concentrations of 9 mg/kg and 6 mg/kg. The Se enrichment coefficient in the above-ground of *ABH* with biochar application was the lowest (1.807 ± 0.263) in the Se concentration of the control treatment. The transport coefficient was the highest in control treatment in same group of *Astragalus* species, while *ABH* with biochar application was the lowest (0.440 ± 0.017) in the Se concentration of 6 mg/kg in every group treatment.

### 2.5. Correlation Analysis between Se and Growth Indexes and Active Components in ABH

The correlation analysis between Se and growth indexes and active components in *ABH* is shown in Table 2. There was a significant positive correlation between Se content in roots and Se content in above-ground, plant height, main root length, and calycosin-7-glucoside content in roots (*p* < 0.01), and the correlation coefficients were 0.955, 0.682, 0.517 and 0.775, respectively, but there was no significant correlation between total flavonoid content and total polysaccharide content in roots (*p* > 0.05).

## 3. Discussion

Se is one such candidate, which though not considered among essential plant elements, has been seen in the recent past to act as a consequential micronutrient for humans and plants [19]. Se cannot be synthesized in the body on its own and is easily metabolized, so people must obtain Se from their diet to prevent Se deficiency [20]. Compared to the inorganic Se added to Se supplementation drugs on the market, the organic Se enriched in plants is more stable and easily absorbed by the body [21]. Another important aspect of Se usage is that it can relieve the plants from damaging effects caused by various abiotic stresses, e.g., drought [22] and high-temperature [23].

Exogenous Se has been recognized as having a positive impact on plant growth and development [19]. Studies have demonstrated that Se application can stimulate growth in rice [24] and maize [25] and enhance the dry weight of tomatoes [26]. A study by Zhao et al. [27] revealed that exogenously supplied Se improved the yield of *Brassica chinensis* L., with higher concentrations of Se fertilizer significantly promoting plant growth. Se-fortified plants have been developed to address deficiencies in both human and livestock populations. Hyperaccumulators are particularly intriguing due to their capacity for organic accumulation of selenium, which is known to exhibit powerful anticarcinogenic properties by achieving exceptionally high levels of concentration [6]. *Astragalus* species of legume that hyperaccumulate Se manage to amass extreme amounts of Se from seleniferous soil. Given the ability of hyperaccumulators to amass trace elements, hyperaccumulation is thought to benefit the plant in some way [6].

In this study, we found that exogenous Se could promote the growth of *Astragalus* species (Figure 1, Figure 2 and Figure 3), similar to the experimental results of Zhao et al. [27]. The maximum value was reached in Se concentration of 6 mg/kg for the plant height and main root length, the rhizome diameter and ground diameter, or the whole plant dry weight and root dry weight. The minimum value was reached in Se concentrations of 0 mg/kg or 12 mg/kg. This result indicated that the medium concentration of Se (6 mg/kg) was the most conducive to the growth of *Astragalus* species. On the other hand, the control (0 mg/kg) and high concentration (12 mg/kg) treatments did not foster optimal growth. Research has demonstrated that selenium (Se) possesses a biphasic effect on plant growth. Specifically, low concentrations of Se have been shown to stimulate plant growth and enhance its antioxidant capacity. Conversely, high Se exposure has been found to suppress plant growth and expedite both plant maturation and senescence [28]. The Se content of *ABH* and *AB* gradually increased with the increase of Se concentrations whether in roots or above-ground (Figure 6 and Figure 7). Individually, the higher the Se levels in the soil, the greater change in the Se concentrations in *Astragalus* species, which further demonstrated that the chemical composition formation of medicinal plants was related to the Se levels in the soil. This may represent an ‘ideal’ concentration of Se in soil for hyperaccumulators, where lower levels induce foraging behavior and higher concentrations allow non-foraging behavior to still result in beneficial levels of Se uptake, especially considering the intensive cycling of Se within the root system.

Biochar has been demonstrated to improve various crucial soil characteristics, including pH, water-holding capacity, cation exchange capacity, nutrient availability, and microbial activities. These factors are all closely associated with the amendment of soil health [14,29]. The applying biochar had a significant effect on reducing soil bulk density, improving the ground temperature, and increasing soil moisture content [30]. Zhang et al. [31] demonstrated that biochar application led to a significant increase in seed yield for *Oryza sativa* L. Furthermore, Chen’s (2020) investigation indicated that the integration of biochar into the soil could enhance the growth potential of *Astragalus* species. Specifically, the addition of biochar promoted increases in plant height, taproot diameter, length, weight, survival rate, and yield.

The addition of biochar (cotton straw charcoal) and a certain concentrations range of Se treatments had a significant promotion effect on the growth of *Astragalus* species (Figure 1, Figure 2 and Figure 3), which was basically consistent with the experimental results of Chen [30]. In general, the application of biochar in the soil led to significant increases in plant height, main root length, rhizome diameter, ground diameter, and both whole plant and root dry weight for *Astragalus* species. These increases were observed in the biochar-treated groups even with identical Se concentration treatments. The Se content of *ABH* with biochar application was higher than those of *ABH* without biochar application in the same Se concentrations whether in roots or above ground (Figure 4 and Figure 5). It is reported that humus as a major soil organic matter in soil can complex with Se, and the organic matter-bound Se is a potential Se source required by plants’ growth [32]. Specifically, humus can promote the reduction of Se in the soil by providing a carbon source to the microorganisms [33,34]. An experimental study on three medicinal plants demonstrated that adding biochar amendments resulted in significant improvements in plant biomass ranging from 21% to 175%. Furthermore, there was an observable enhancement in secondary metabolite content as a direct result of the biochar application [18]. In this study, we found that biochar can not only promote the growth of *Astragalus* species and the uptake of exogenous Se but also increase the active component contents (Figure 8, Figure 9 and Figure 10). The contents of total flavonoid, total polysaccharide, and calycosin-7-glucoside in *ABH* roots with biochar application were higher than in *ABH* without biochar application. However, the effective component contents in *ABH* roots increased first and then decreased with the increase of Se concentrations, in other words, biochar application in higher Se concentration was not conducive to accumulating active components in *Astragalus* species. The current study observed that variations in Se levels within soil appeared to have an impact on the formation and distribution of different constituents of humus. Evidence suggests that excess Se concentrations may interfere with biochar adsorption, further restricting the formation of active ingredients.

Plant species differ in their ability to take up and assimilate Se [10]. Differences in root morphology and biochemical or physiological pathways are associated with different uptake rates. There is still a considerable gap in scientific understanding of the below-ground mechanisms that facilitate the ability of hyperaccumulators to accumulate significant quantities of trace elements, including selenium (Se), directly from the soil [6]. The Se enrichment coefficients of *Astragalus* species in Se treatments were higher than that in the control treatment and there were significant differences (*p* < 0.05) in all treatments (Table 1). The Se transport coefficients of *Astragalus* species were less than 1, except for the transport coefficient of *AB* at the Se concentration of the control treatment (0 mg/kg) (Table 1). This indicated that the absorption and transport capacity of Se in the roots of *Astragalus* species was stronger than that in above-ground portions, and biochar was more conducive to the absorption and transport of Se in the roots of *Astragalus* species but not in the above-ground portions. Selenate is easily leached to deeper layers in the soil profile [35]. Volatilization is an important mechanism of Se loss. Plant species, soil characteristics, and weather conditions all affect the rate of volatilization [10]. The correlation analysis between Se and growth indexes and active components in *ABH* also well proves this point.

To sum up, the present study demonstrates biochar amendments not only promote plant growth but also improve exogenous Se uptake and the metabolite content in *Astragalus* species even higher than the control treatment. Therefore, the differences in Se levels in soil may have two effects. First, it may affect the structure of the biochar components or the content of active groups, which in turn affected the adsorption of Se by biochar components. Second, it may affect the combination of Se and biochar on active component formation.

## 4. Materials and Methods

### 4.1. Plant Material and Soil Physicochemical Property Measurement

According to Pharmacopoeia of People’s Republic of China (2020), *A. membrauceus* (Fisch.) Bge. var. *mougholicus* (Bge.) Hsiao (*ABH*) and *A. membrauaceus* (Fisch.) Bge. (*AB*) were used in the experiment. The tested selenium (Se) was sodium selenite (Na_2_SeO_3_) produced by Tianjin Chemical Reagent Factory. The tested biochar as humus was cotton straw biochar provided by the Soil Fertilizer and Water Saving Research Institute of Xinjiang Academy of Agricultural Sciences in China. The soil samples were collected from the surface soil layer (0–20 cm) of Anningqu field in Urumqi of Xinjiang in China, the basic physical and chemical properties of the soil were as follows: pH 7.98, total salt 2.4 g/kg, organic matter 3.29 g/kg, available nitrogen 48.0 mg/kg, available phosphorus 4.4 mg/kg, available potassium 172.0 mg/kg, and total selenium (Se) 0.146 mg/kg. The soil was air dried, ground, and sifted through 2 mm sieve.

### 4.2. Experimental Design

The soil pot experiment was conducted to explore the effects of different concentrations of sodium selenite (SeⅣ) added in soil on Se uptake, translocation, and accumulation in *Astragalus* species, according to the method of Geng et al. [36]. Three factors were investigated including plant species (*ABH* and *AB*), Se concentrations (0, 3, 6, 9, and 12 mg/kg), and soil condition (with and without biochar application), a total of 15 treatments were tested. The grouping information was shown in Table 3, and each treatment was repeated four times.

The specific method was as follows: chosen 22.5 × 23.5 cm plastic basin, each basin was filled with soil 7 kg to each was added 7.68 g phosphorus and potassium fertilizer (monopotassium), 1.50 g nitrogen fertilizer (urea), and different concentrations of Se were sprayed into soil samples with sprayers. According to Ge et al. ’s method [37], 39 g cotton straw biochar was added to the biochar group of *ABH*, which was fully mixed and then loaded into the basin. Six seeds of *Astragalus* species were sown in each pot, and the sowing depth was 1.0–1.5 cm. The sowing time was on 1 April 2020. After sprouting, 4 plants of them were planted in each pot. The plants had grown under natural light and had been watered with 1 L of distilled water every three days regularly. After 240 days of growth, the above-ground portions and roots of *Astragalus* species were harvested to determine the growth indexes, Se content, and active component content.

### 4.3. Determination of Plant Growth Indexes

The harvested *Astragalus* plants were first washed with tap water, followed by rinsing with distilled water, and then blotted dry with absorbent paper. The above-ground and underground portions of the plants were subsequently separated, and measurements were taken for plant height (distance from stem base to plant top), main root length (the longest distance from rhizome to main root), ground diameter (diameter 0.5 cm upward from the base of the main stem), and rhizome diameter (root diameter at rhizome). All plant parts were then dried at 60 °C until a constant dry mass was achieved. The dry weight of both the whole plant and roots was recorded.

### 4.4. Determination of Active Components and Se Content

The total flavonoid content was determined by UV spectrophotometry according to Liu’s method [38]. The total polysaccharides content was determined by the phenol sulfuric acid method according to Yang et al.’s method [39]. Additionally, the calycosin-7-glucoside content was determined by high-performance liquid chromatography (HPLC) according to Wu et al.’s method [40]. According to Li et al.’s method [41], plant samples and soil samples were digested with HNO_3_-HClO_4_ (volume ratio of 4:1), and Se content was determined by hydride generation atomic fluorescence spectrometry (HG-AFS).

The relevant standard curves and linear relationships were shown in Figure 11 and Table 4. The enrichment coefficient refers to the ratio of Se content above ground and roots of *Astragalus* species to Se content in the soil. The transport coefficient refers to the ratio of Se content above ground to Se content in roots.

### 4.5. Data Analysis

Differences in plant height and main root length, rhizome diameter and ground diameter, and the dry weight of the whole plant and root were assessed through two-way ANOVA and Fisher LSD post hoc test considering species (*ABH* and *AB*), Se concentration treatments (0, 3, 6, 9 and 12 mg/kg) and soil condition (with and without biochar application) as factors. Differences in the Se content in *ABH* roots and *ABH* above-ground portions were assessed through two-way ANOVA and Fisher LSD post hoc test considering Se concentration treatments and soil condition as factors. Differences in the Se content in roots and in above-ground portions were assessed through two-way ANOVA and Fisher LSD post hoc test considering species and Se concentration treatments as factors. Differences in total flavonoid content, total polysaccharide content, and calycosin-7-glucoside content in *ABH* roots were assessed through two-way ANOVA and Fisher LSD post hoc test considering Se concentration treatments and soil condition as factors. All statistical tests were performed with SPSS version 22.0. considering a significant level of *p* < 0.05 or *p* < 0.01.

## 5. Conclusions

The integration of a medium concentration of Se treatments, in combination with the addition of biochar, has been observed to stimulate growth and promote Se content, as well as support the accumulation of active components in *Astragalus* species. Moreover, this combined treatment approach can augment both the Se enrichment and transport capacity within the plant. The Se enrichment coefficients of *Astragalus* species in Se treatments were higher than that in control treatment and there were significant differences whether roots or above-ground portions in all treatments. The Se transport coefficients of *Astragalus* species were less than 1, except for the transport coefficient of *AB* in the Se concentration of the control treatment (0 mg/kg). This indicated that the absorption and transport capacity of Se in the roots of *Astragalus* species was stronger than that in above-ground portions, and biochar application was more conducive to the absorption and transport of Se in the root of *Astragalus* species but not in the above-ground portions. The correlation analysis between Se and growth indexes and active components in *ABH* also well proves this point. The contents of Se and active components in roots of *Astragalus* species are closely related to germplasm, soil substrate, and applied exogenous Se concentrations. Therefore, the influence of *Astragalus* species, soil substrate, and applied exogenous Se concentrations on the quality of *Astragalus* species should be fully considered during the cultivation of Se-rich *Astragalus* species. Taken as a whole, the present findings offer a more comprehensive understanding of the interplay between distinct Se levels, as well as the addition of biochar in soil, providing valuable insight for the cultivation of Se-rich *Astragalus* in Se-deficient soil-plant systems.

## Figures and Tables

**Figure 1 plants-12-01957-f001:**
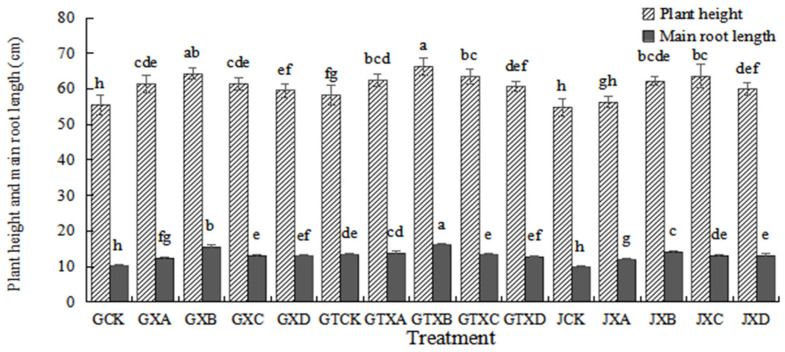
Plant height and main root length of *Astragalus* species in different treatments. Values are mean ± SEM (*n* = 4). Different letters show statistical differences using two-way ANOVA considering species (*ABH* or *AB*), Se concentration treatments (0, 3, 6, 9, and 12 mg/kg), and soil condition (with or without biochar) as factors (Fisher LSD test; *p* < 0.05). Note: GCK, GXA, GXB, GXC, and GXD represented different Se treatment levels 0, 3, 6, 9 and 12 of *Astragalus membrauceus* (Fisch.) Bge. var. *mongholicus* (Bge.) Hsiao (*ABH*), respectively. GTCK, GTXA, GTXB, GTXC, and GTXD represented different Se treatment levels 0, 3, 6, 9, and 12 of *ABH* with biochar, respectively. JCK, JXA, JXB, JXC, and JXD represented different Se treatment levels 0, 3, 6, 9, and 12 of *A. membrauaceus* (Fisch.) Bge (*AB*), respectively.

**Figure 2 plants-12-01957-f002:**
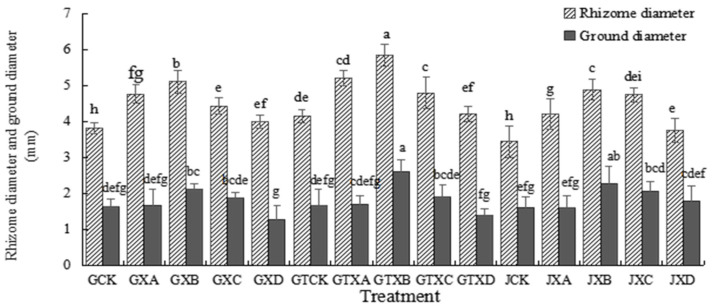
Rhizome diameter and ground diameter of *Astragalus* species in different treatments. Values are mean ± SEM (*n* = 4). Different letters show statistical differences using two-way ANOVA considering species (*ABH* or *AB*), Se concentration treatments (0, 3, 6, 9, and 12 mg/kg), and soil condition (with or without biochar) as factors (Fisher LSD test; *p* < 0.05). Note: GCK, GXA, GXB, GXC, and GXD represented different Se treatment levels 0, 3, 6, 9 and 12 of *Astragalus membrauceus* (Fisch.) Bge. var. *mongholicus* (Bge.) Hsiao (*ABH*), respectively. GTCK, GTXA, GTXB, GTXC, and GTXD represented different Se treatment levels 0, 3, 6, 9, and 12 of *ABH* with biochar, respectively. JCK, JXA, JXB, JXC, and JXD represented different Se treatment levels 0, 3, 6, 9, and 12 of *A. membrauaceus* (Fisch.) Bge (*AB*), respectively.

**Figure 3 plants-12-01957-f003:**
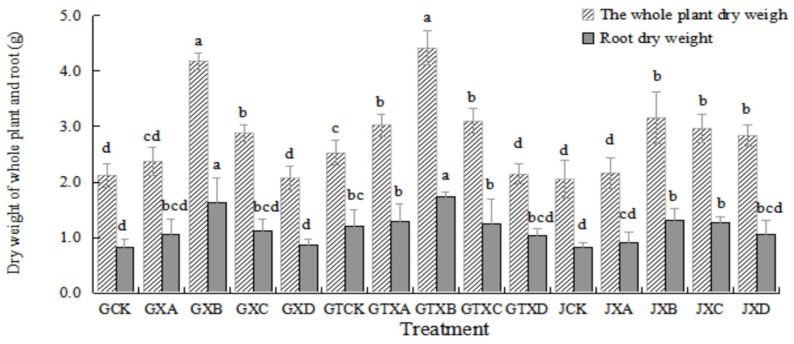
Dry weight of whole plant and root of *Astragalus* species in different treatments. Values are mean ± SEM (*n* = 4). Different letters show statistical differences using two-way ANOVA considering species (*ABH* or *AB*), Se concentration treatments (0, 3, 6, 9, and 12 mg/kg) and soil condition (with or without biochar) as factors (Fisher LSD test; *p* < 0.05). Note: GCK, GXA, GXB, GXC and GXD represented different Se treatment levels 0, 3, 6, 9, and 12 of *Astragalus membrauceus* (Fisch.) Bge. var. *mongholicus* (Bge.) Hsiao (*ABH*), respectively. GTCK, GTXA, GTXB, GTXC and GTXD represented different Se treatment levels 0, 3, 6, 9 and 12 of *ABH* with biochar, respectively. JCK, JXA, JXB, JXC, and JXD represented different Se treatment levels 0, 3, 6, 9, and 12 of *A. membrauaceus* (Fisch.) Bge (*AB*), respectively.

**Figure 4 plants-12-01957-f004:**
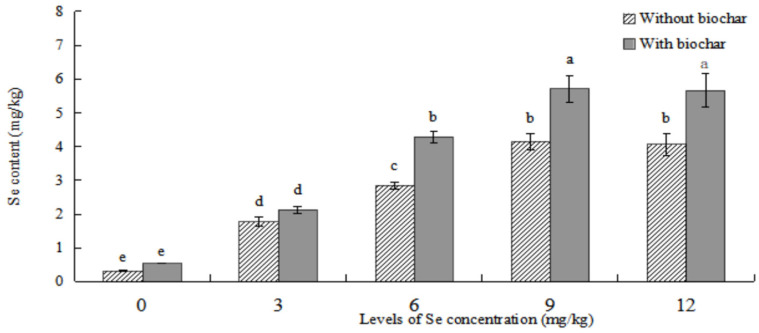
Effects of biochar application on Se content in *ABH* roots. Values are mean ± SEM (*n* = 4). Different letters show statistical differences using two-way ANOVA considering levels of Se concentration treatments (0, 3, 6, 9, and 12 mg/kg) and soil condition (with or without biochar) as factors (Fisher LSD test; *p* < 0.05).

**Figure 5 plants-12-01957-f005:**
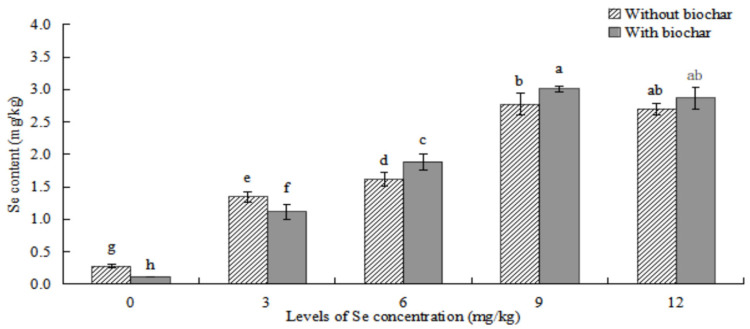
Effects of biochar application on Se content in *ABH* above ground. Values are mean ± SEM (*n* = 4). Different letters show statistical differences using two-way ANOVA considering levels of Se concentration treatments (0, 3, 6, 9, and 12 mg/kg) and soil condition (with or without biochar) as factors (Fisher LSD test; *p* < 0.05).

**Figure 6 plants-12-01957-f006:**
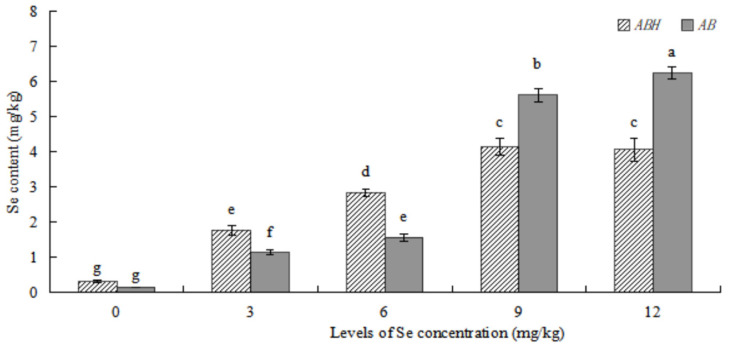
Comparison of Se content in the roots of *ABH* and *AB*. Values are mean ± SEM (*n* = 4). Different letters show statistical differences using two-way ANOVA considering species (*ABH* or *AB*) and levels of Se concentration treatments (0, 3, 6, 9, and 12 mg/kg) as factors (Fisher LSD test; *p* < 0.05).

**Figure 7 plants-12-01957-f007:**
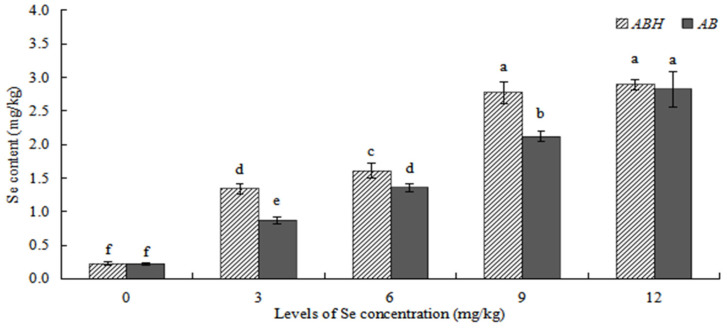
Comparison of Se content in the above-ground of *ABH* and *AB*. Values are mean ± SEM (*n* = 4). Different letters show statistical differences using two-way ANOVA considering species (*ABH* or *AB*) and levels of Se concentration treatments (0, 3, 6, 9, and 12 mg/kg) as factors (Fisher LSD test; *p* < 0.05).

**Figure 8 plants-12-01957-f008:**
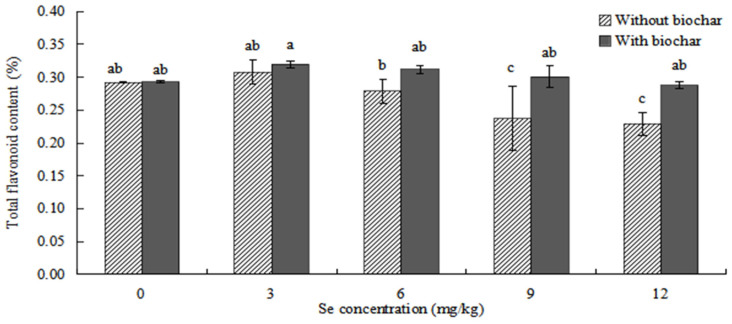
Effects of biochar application on total flavonoid content in *ABH* roots. Values are mean ± SEM (*n* = 4). Different letters show statistical differences using two-way ANOVA considering levels of Se concentration treatments (0, 3, 6, 9, and 12 mg/kg) and soil condition (with or without biochar) as factors (Fisher LSD test; *p* < 0.05).

**Figure 9 plants-12-01957-f009:**
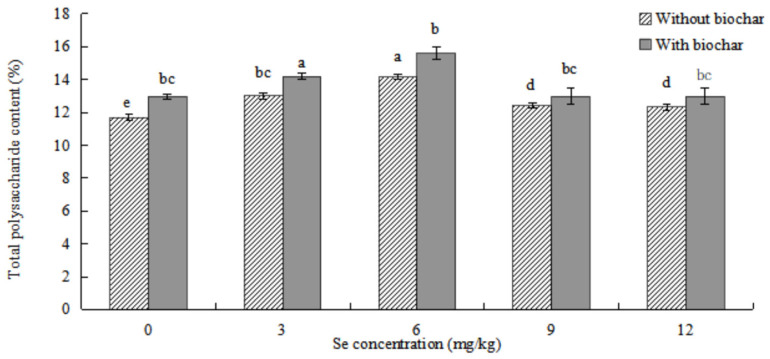
Effects of biochar application on total polysaccharide content in *ABH* roots. Values are mean ± SEM (*n* = 4). Different letters show statistical differences using two-way ANOVA considering levels of Se concentration treatments (0, 3, 6, 9, and 12 mg/kg) and soil condition (with or without biochar) as factors (Fisher LSD test; *p* < 0.05).

**Figure 10 plants-12-01957-f010:**
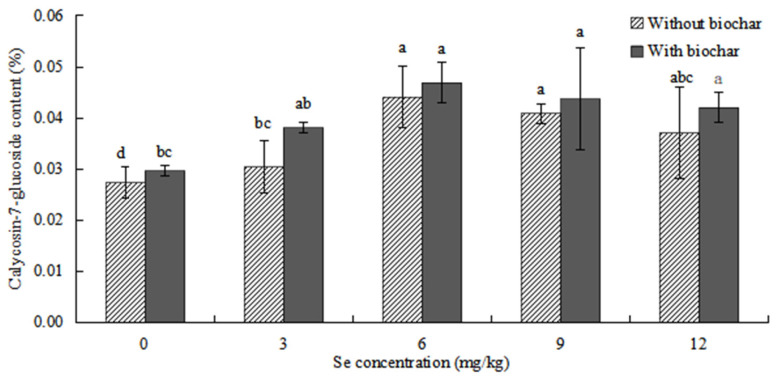
Effects of biochar application on calycosin-7-glucoside content in *ABH* roots. Values are mean ± SEM (*n* = 4). Different letters show statistical differences using two-way ANOVA considering levels of Se concentration treatments (0, 3, 6, 9, and 12 mg/kg) and soil condition (with or without biochar) as factors (Fisher LSD test; *p* < 0.05).

**Figure 11 plants-12-01957-f011:**
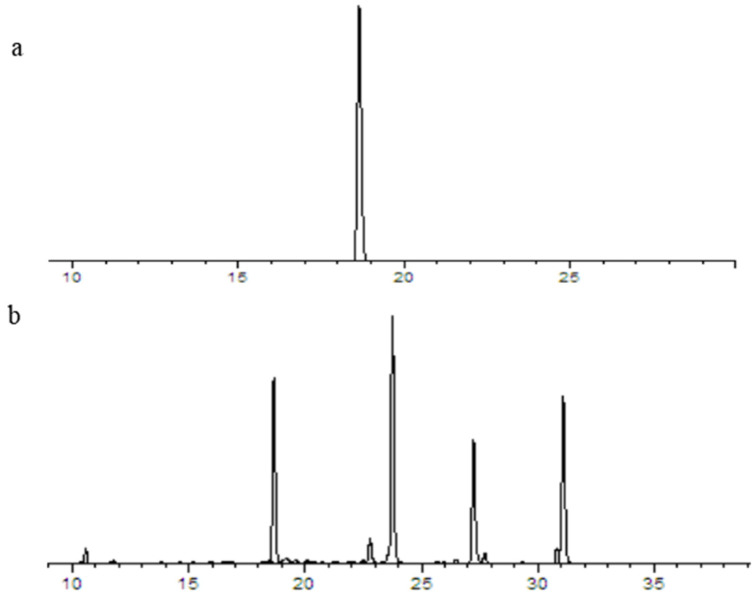
Chromatogram of reference substance (**a**) and sample (**b**) of calycosin-7-glucoside.

**Table 1 plants-12-01957-t001:** Effects of different treatments on the enrichment and transport characteristics of Se in the roots and above-ground of *Astragalus* species.

Group Information	Se Concentration (mg/kg)	Enrichment Coefficient in Roots	Enrichment Coefficient in Above-Ground	Transport Coefficient
*ABH*	0	2.463 ± 0.187 h	2.158 ± 0.238 i	0.875 ± 0.041 b
	3	6.744 ± 0.334 f	5.124 ± 0.145 d	0.757 ± 0.031 c
	6	9.169 ± 1.032 e	5.158 ± 0.215 d	0.565 ± 0.018 g
	9	6.522 ± 0.524 f	4.349 ± 0.032 ef	0.668 ± 0.010 e
	12	5.872 ± 0.516f g	4.176 ± 0.064 f	0.714 ± 0.037 d
*ABH* with biochar	0	8.414 ± 1.602 e	1.807 ± 0.263 d	0.620 ± 0.022 f
	3	12.323 ± 1.042 d	6.446 ± 0.583 c	0.524 ± 0.045 gh
	6	19.468 ± 2.557 a	8.516 ± 0.302 a	0.440 ± 0.017 i
	9	16.653 ± 1.592 b	8.768 ± 0.251 a	0.529 ± 0.031 gh
	12	14.708 ± 1.390 c	7.455 ± 0.665 b	0.509 ± 0.054 h
*AB*	0	1.934 ± 0.190 h	2.896 ± 0.232 h	1.500 ± 0.046 a
	3	4.752 ± 0.995 g	3.633 ± 0.456 g	0.774 ± 0.027 c
	6	4.420 ± 0.159 g	3.397 ± 0.084 e	0.769 ± 0.032 c
	9	9.980 ± 0.113 d	4.523 ± 0.111 ef	0.453 ± 0.009 j
	12	10.756 ± 0.400 d	4.847 ± 0.294 de	0.451 ± 0.031 i

Values are mean ± SEM (*n* = 4). Different letters show statistical differences using two-way ANOVA considering species (ABH or AB), Se concentration treatments (0, 3, 6, 9, and 12 mg/kg), and soil condition (with or without biochar) as factors (Fisher LSD test; *p* < 0.05).

**Table 2 plants-12-01957-t002:** Correlation analysis results in *ABH*.

Components	Se Content in Roots	Se Content in Above-Ground	Plant Height	Taproot Length	Total Flavonoid	Total Polysaccharide	Calycosin-7-Glucoside
Se content in roots	1.000	0.955 **	0.682 **	0.517 **	−0.152	0.367	0.775 **
Se content in above-ground		1.000	0.581 **	0.352	−0.286	0.193	0.659 **
Taproot length			1.000	0.769 **	0.101	0.694 **	0.653 **
Plant height				1.000	0.105	0.901 **	0.685 **
Total flavonoid					1.000	0.224	−0.051
Total polysaccharide						1.000	0.626 **
Calycosin-7-glucoside							1.000

The ** after the numbers shows extremely significant difference at *p* < 0.01.

**Table 3 plants-12-01957-t003:** Group information.

Se Concentration (mg/kg)	*ABH*	*ABH* with Biochar	*AB*
0	GCK	GTCK	JCK
3	GXA	GTXA	JXA
6	GXB	GTXB	JXB
9	GXC	GTXC	JXC
12	GXD	GTXD	JXD

**Table 4 plants-12-01957-t004:** Investigation result of linear relationship.

Component	Linear Relationship	R^2^	Linear Range
Total flavone	Y_1_ = 71.415X_1_ + 0.0178	0.9998	2.56–25.60 μg/mL
Polysaccharide	Y_2_ = 10.73X_2_ − 0.065	0.9991	20.00–100.00 μg/mL
Calycosin-7-glucoside	Y_3_ = 1,651,116.4080X_3_ − 2740.1121	0.9999	0.04–0.60 μg
Se	Y_4_ = 167.63X_4_ + 0.0007	0.9999	1.00–20.00 μg/L

## Data Availability

The datasets used and/or analyzed during the current study are available from the corresponding author on reasonable request.

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
