# Peer review of "Exogenous Selenium and Biochar Application Modulate the Growth and Selenium Uptake of Medicinal Legume Astragalus Species"

_plants, 2023, doi:10.3390/plants12101957_

Round 1

Reviewer 1 Report

The manuscript by ma et al. reports investigations on how exogenous selenium and biochar application could modulate the growth and selenium uptake 2 of medicinal Legume Astragalus species.

Even though the manuscript's content is interesting and the results should be published. The manuscript could not be accepted in the present form. To my opinion, the authors submitted an advanced draft but not the final paper (at line 5.15 of the pdf file, a green highlighter is present).

Other critical comments are:

1. At specific points, the text is tough to read (such as lines 462- 466). Thus I suggest a careful English revision.

2. Please use and carefully follow the Plants template: the Abstract should be only 200 words (the present Abstract is more than 500 words). The headers of the sub-chapters should start with numbers or at least be separated from the other text. This could increase the readability of the manuscript.

3. All the names of species should be in italics. They are in italics only in the captions of the figures. 

4. In the introduction. The authors should justify why they decided to study flavonoids, sugars and calycosin-7-glucoside and why studying this compound in the legume Astragalus species is essential.

5. Methods. All the methods used should be reported in total. It is not advisable to cite the reference as the authors did in the " Determination of active components and Se content" section. On the other hand, the information about the methods should be complete to help the transparency. Moreover, Figure 1 should be moved in the results. Furthermore, considering that the authors used different treatments and species, it could be essential to report a chromatogram x treatment per species as supplementary figures. Finally, in line 160, it is unclear where "the enrichment coefficient" and "transport coefficient" come out, probably from a method not wholly stated.

5. Results and Discussion. Table 4 should be moved from the Discussion to the results after the first citation.

Reviewer 2 Report

Dear Authors,

Your manuscript titled 'Exogenous selenium and biochar application modulate the growth and selenium uptake of medicinal Legume Astragalus species' is addressing an interesting topic of research for cultivating Se-rich Astragalus species and improving soil conditions. However, it needs major revision. Please, see below a few comments and suggestions to be applied by you before acceptation of your manuscript for publication at Plants MDPI.

L3 Replace 'Legume' by 'legume'.

L15 Replace 'Legume' by 'legume'.

L20  Replace 'Legume' by 'legume'.

L23-31 Justify by edition all this paragraph.

L24 Replace 'biocharwere' by 'biochar were'.

L25 Replace 'biochar' by 'biochar, respectively'.

L25-L26 Delete 'With the way of soil cultured. in pots in the experiment.'.

L33-L46 Justify all this paragraph by edition.

L36-L37 Replace 'bi-ochar' by 'bio-char'.

L48-L50 Justify all this paragraph by edition.

L54 Delete 'Enrich selenium (Se)' and 'Astragalus species'.

L54 Add 'biochar application' and 'hyperaccumulator legumes'.

L59 Replace 'Commis-sion' by 'Commi-ssion'.

L63 Replace 'within' by 'from' and 'Legume' by 'legumes'.

L64 Replace 'has' by 'have'.

L65 Replace 'difference' by ''differences'.

L98 Replace 'Legume' by 'legumes'. 

L100 Replace 'We hope the results and methods could' by 'We expect that our results and methods applied herein could'.

L56-L102 Please, review carefully the Introduction section and try to justify why the three factors under study are relevant for testing 2 Astragalus species (ABH vs. AB), 5 levels of Se (0, 3, 6, 9 and 12 mg/kg) and 2 soil conditions (with vs. without biochar application).

L56-L102 Adapt all references cited herein to MDPI's guidelines for authors.

L104 Distinguish this subsection following MDPI's guidelines for authors.

L108 Write the numbers with subscript.

L116 Distinguish this subsection following MDPI's guidelines for authors.

L118 Delete (0,3, 6, 9 and 12 mg/kg).

L119 Replace 'Astragalus' by 'Astragalus species'.

L120 Replace 'There were three factors' by 'Three factors were investigated'.

L121-122 'a total of 15 treatments.' by ' and a total of 15 treatments were tested'.

L120-L123 Please, clarify how are testing only 15 treatments? According to your research desing you are investigating 2 Astragalus species (ABH vs. AB), 5 levels of Se (0, 3, 6, 9 and 12 mg/kg) and 2 soil conditions (with vs. without biochar application) so you should have tested 2*5*2= 20 treatments. Why are you not describing the 5 treatments left at Table 1 from AB with biochar? Verify the experimental design and justify the acronyms for your treatments.

L127 Replace 'wasfilled' by 'was filled' and ''7 kg, each' by '7 kg to each were'.

L128 Replace '(urea),' by '(urea), and'.

L129 Justify this line by edition.

L137 Distinguish this subsection following MDPI's guidelines for authors.

L143 Replace '60 ºC' by  '60ºC'.

L144 Replace 're-spectively' by 'res-pectively'.

L146 Distinguish this subsection following MDPI's guidelines for authors.

L156-158 exchange the position of Figure 1 by Table 2. Move them.

L163 Distinguish this subsection following MDPI's guidelines for authors.

L169 Replace 'consid-ering' by 'consi-dering'.

L176 Replace 'signif-icant' by 'signi-ficant'.

L179 Distinguish this subsection following MDPI's guidelines for authors.

L180 Replace 're-sponse' by 'res-ponse'.

L189 Replace 'respec-tively' by 'respe-ctively'.

L192 Replace 'were highest' by 'were the highest'.

L192 Replace 'dif-ferent' by 'di-fferent'.

L193 Replace ', and' by ' while'.

L194 Replace 'were lowest' by 'were the lowest'.

L194-L195 Delete 'The former ... , respectively.'.

Figure 2: Delete the decimals on the y-axis.

L198 Replace 'SE' by 'standard error of the mean (SEM)'.

L200 Add reference to each treatment as Figure 2 subnote.

L208 Replace 'diam-eter' by 'diame-ter'.

L209 Replace 'were highest' by 'were the highest'.

L211 Replace 'were lowest' by 'were the lowest'.

L211-L212 Delete 'The former ... , respectively.'.

Figure 3: Delete the decimals on the y-axis.

L215 Replace 'SE' by 'SEM'.

L226 Replace 'ap-plication' by 'appli-cation'.

L227-L228 Replace 'were highest' by 'were the highest'.

L229 Replace 'were lowest' by 'were the lowest'.

L230 Delete 'The former ... , respectively.'.

Figure 4: Delete the decimals on the y-axis and replace 'The dry' by 'Dry'.

L233 Replace 'SE' by 'SEM'.

L236 Distinguish this subsection following MDPI's guidelines for authors.

L237 Distinguish also it following MDPI's guidelines for authors.

L240 Replace 'ap-plication' by 'appli-cation'.

L245 Replace 'respective treatments' by 'respective to treatments'.

L247 Replace 'was lowest' by ''was the lowest'.

L248 Delete 'The former ...'.

Figure 5: Delete the decimals on the y-axis and replace 'Se concentration' by 'levels of Se concentration'.

L250 Replace 'SE' by 'SEM'.

L254 Distinguish this subsection following MDPI's guidelines for authors.

L260 Replace 'was highest' by 'was the highest'.

L261-L262 Delete 'The former ...'.

Figure 6: Delete the decimals on the y-axis and replace the scale from 0 to 4. Also, replace 'Se concentration' by 'levels of Se concentration' on the x-axis.

L264 Replace 'SE' by 'SEM'.

L268 Distinguish this subsection following MDPI's guidelines for authors.

L274 Replace '(0.320)' by '(0.320 mg/kg)'.

L275-L276 Delete 'The Se ...'.

L278-L279 Delete 'The former ...'.

Figure 7: Delete the decimals on the y-axis and replace 'Se concentration' by 'levels of Se concentration' on the x-axis.

L281 Replace 'SE' by 'SEM'.

L281-L282 Replace 'Dif-ferent' by 'Diffe-rent'.

L284 Distinguish this subsection following MDPI's guidelines for authors.

L290 Replace 'mg/kg, 0.224' by 'mg/kg, 0.224', '0.014 mg/kg' by '0.014 mg/kg, respectively' and 'There was' by 'There were'.

L291 Replace 'difference' by 'differences'.

L292 Replace 'was' by 'were' and 'difference' by 'differences'.

L293-L294 Delete 'The Se ...'.

Figure 8: Delete the decimals on the y-axis and replace 'Se concentration' by 'levels of Se concentration' on the x-axis.

L296 Replace 'SE' by 'SEM'.

L299 Distinguish this subsection following MDPI's guidelines for authors.

L300 Distinguish also it following MDPI's guidelines for authors.

L306 Replace '0.018 %,' by '0.018% and'  and '0.005 %' by '0.005%, respectively'.

L307 Replace 'whether' by 'whether it is'.

L308 Replace '0.017 %,' by '0.017% and' and '0.006 %' by '0.006%, respectively'.

L308-L309 Delet 'The former ...'.

Figure 9: Replace 'Se concentration' by 'levels of Se concentration' on the x-axis.

L312 Replace 'SE' by 'SEM'.

L315 Distinguish this subsection following MDPI's guidelines for authors.

L321 Replace 'were highest' by 'were the highest' and '0.159 %,' by '0.159% and' and '0.366 %' by '0.366%, respectively'.

L322 Replace 'without,' by 'without, respectively'.

L324 Delete 'The former ...'.

Figure 10: Delete the decimals on the y-axis and replace 'Se concentration' by 'levels of Se concentration' on the x-axis.

L327 Replace 'SE' by 'SEM'.

L331 Distinguish this subsection following MDPI's guidelines for authors.

L336 Replace 'difference' by 'differences' and 'concentration' by 'concentrations'.

L338 Replace '0.006 %,' by '0.006% and' and '0.004 %' by '0.004%, respectively'.

L339 Replace 'without,' by 'without, respectively' and '0.003 %,' by '0.003% and' and '0.001 %' by '0.001%, respectively'.

L340-L341 Delete 'The former ...'.

Figure 11: Replace 'Se concentration' by 'levels of Se concentration' on the x-axis.

L344 Replace 'SE' by 'SEM'.

L348 Distinguish this subsection following MDPI's guidelines for authors.

L353 Replace 'difference' by 'differences'.

L356 Replace 'Effects' by 'Effect', 'ABH' by 'ABH without biochar' and 'AB' by 'AB without biochar'.

L358 Replace 'SE' by 'SEM'.

L358-L360 Justify this paragraph by edition.

L365 Replace 'bi-ochar' by 'bio-char'.

L368 Replace '2.557' by '2.557 mg/kg, respectively'.

L369 Replace '0.187,' by '0.187 and' and '1.602' by '1.602 mg/kg, respectively'.

L369 Delete 'The former ...'.

L376 Replace '0.251' by '0.251 mg/kg' and delete a space before 'followed'.

L377 Replace 'was' by 'were' and 'difference' by 'differences'.

L379 Replace '0.263' by '0.263 mg/kg'.

L380 Delete 'The former ...'

L380-L381 Replace 'coef-ficient' by ''coeffi-cient'.

L381 Replace 'which of' by 'while'.

L384 Distinguish this subsection following MDPI's guidelines for authors.

L390 Insert Table 4 here. Move it from Discussion section.

L392 Replace 'el-ements' by 'ele-ments'.

L392-L432 Review Discussion. Try to be clair in your statements and not too much diffuse. Compare your results with previous work done by others considering your factors under study. Please, be concise and well structured.

L404 Write in italics 'Brassica chinensis'.

L410 Replace 'Legume' by 'legumes'.

L416 Replace 'whether' by 'for'.

L435 Replace 'bi-ochar' by 'bio-char'.

L439 Replace '(2020) study found' by '(2020) study also found.

L456 Replace 'bi-ochar' by 'bio-char'.

L459 Replace 'ap-plication' by 'appli-cation'.

L469 Replace 'associ-ated' by 'associa-ted'.

L470 Replace 'belowground' by 'below ground'.

L474 Replace 'was' by 'were' and 'difference' by 'differences'.

L480 Replace 'mech-anism' by 'mecha-nism'.

L484-485 Move Table 4 to Results section.

L485 Justify this line by edition and replace '0.01' by '0.01.'.

L487 Replace 'In short,' by 'To sum up,'.

L489 Delete 'of Legume'.

L496 Delete 'of Legume'.

L496-L497 Replace 'co-efficients' by 'coe-fficients'.

L499 Delete 'p < 0.05)'.

L495-513 Review Conclusion. Try to be more concise. Avoid present results.

L515 Delete ';'.

L516 Delete 'of some articles'.

L518 Replace 'Conceptual-ization' by 'Conceptuali-zation'.

L527-L651 Review references following MDPI's guidelines for authors. Please, try to cite all of them according to the instructions given the editorial.

L532 Write in italics 'Camelina sinensis'.

L555 Justify this line by edition.

L577 Write in italics 'Neptunia amplexicaulis' and 'Neptunia gracilis'.

L583 Write in italics 'Zea mays'.

L588 Write in italics 'Dendranthema morifolium'.

L597 Replace 'fac-tors' by 'fa-ctors'.

L623 Write in italics 'Brassica juncea'.

L626 Write in italics 'Astragalus mongholicus'.

L639-L640 Write in italics 'Astragalus mongholicus' and 'Astragalus membranaceus'.

Sincerely,

Reviewer.

Round 2

Reviewer 1 Report

The authors responded to the criticisms and modify the manuscript according to the suggestions

Author Response

Thank you very much for your revision, reply and recognition  in your busy schedule, which will certainly encourage me to continue to better engage in scientific research work.

Reviewer 2 Report

Dear Authors,

The manuscript titled "Exogenous selenium and biochard application modulate the growth and selenium uptake of medicinal legume Astragalus species' has been improved after authors' review. It is now acceptable for publication at Plants after minor revision. Please, see below a list of comments to be implemented by you before accepting it for publication.

L21-L22 Replace 'second-ary' by 'seconda-ry'.

L57-L58 Replace 'antiox-idation' by 'antioxi-dation'.

L75-L76 Replace 'sec-ondary' by 'secon-dary'.

L84-L85 Replace 'ap-plied' by 'a-pplied'.

L96-L97 Replace 'dif-ferences' by 'di-fferences'.

L97-L98 Replace 'dif-ferences' by 'di-fferences'.

L121-L122 Replace 'ap-plication' by 'appli-cation'.

L122-L123 Replace 'sim-ilar' by 'si-milar'.

L135-L136 Replace 'Val-ues' by 'Va-lues'.

L152-L153 Replace 'demon-strated' by 'demons-trated'.

L261-L262 Replace 'Astrag-alus' by 'Astra-galus'.

L264-L265 Replace 'coef-ficient' by 'coe-fficient'.

L271-L272 Replace 'respec-tively' by 'respe-ctively'.

L315-L316 Replace 'pow-erful' by 'po-werful'.

L336-L337 Replace 'hyperac-cumulators' by 'hypera-ccumulators'.

L337-L338 Replace 'al-low' by 'a-llow'.

L466-L467 Replace 'di-ameter' by 'dia-meter'.

L498-L499 Replace 'ad-dition' by 'a-ddition'.

L513-L514 Replace 'man-uscript' by 'manus-cript'.

L521-L522 Replace 'Astrag-alus' by 'Astra-galus'.

L539-L540 Replace 'fac-tors' by 'fa-ctors'.

L557-L558 Replace 'Temper-ature' by 'Tempera-ture'.

L558-L559 Replace 'phys-ico' by 'physi-co'.

L570-L571 Replace 'medic-inal' by 'medi-cinal'.

L575-L576 Replace 'mod-esta' by 'mo-desta'.

Kind regards,

Reviewer.

Author Response

Please see the attachment. It should be noted here that  there are some spelling of words in the correct format of the version I submitted at last, so there is no modification.
